

Atmospheric
Chemistry
and Physics

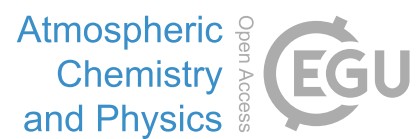

# Particle-bound reactive oxygen species (PB-ROS) emissions and formation pathways in residential wood smoke under different combustion and aging conditions<sub>CE1</sub>

**Jun Zhou**[1], **Peter Zotter**[2], **Emily A. Bruns**[1], **Giulia Stefenelli**[1], **Deepika Bhattu**[1], **Samuel Brown**[1,3], **Amelie Bertrand**[1,4],
**Nicolas Marchand**[4], **Houssni Lamkaddam**[1], **Jay G. Slowik**[1], **André S. H. Prévôt**[1], **Urs Baltensperger**[1],
**Thomas Nussbaumer**[2], **Imad El-Haddad**[1], and **Josef Dommen**[1]

[1]Laboratory of Atmospheric Chemistry, Paul Scherrer Institute, 5232 Villigen PSI, Switzerland
[2]Lucerne University of Applied Sciences and Arts, School of Engineering and Architecture,
Bioenergy Research, 6048 Horw, Switzerland
[3]Institute for Atmospheric and Climate Science, ETH, Zurich, Switzerland
[4]Aix Marseille Univ, CNRS, LCE, Marseille, France

**Correspondence:** Josef Dommen (josef.dommen@psi.ch)

**Abstract.** TS1 TS2 Wood combustion emissions can induce oxidative stress in the human respiratory tract by reactive oxygen species (ROS) in the aerosol particles, which are emitted either directly or formed through oxidation in the atmosphere. To improve our understanding of the particle-bound ROS (PB-ROS) generation potential of wood combustion emissions, a suite of smog chamber (SC) and potential aerosol mass (PAM) chamber experiments were conducted under well-determined conditions for different combustion devices and technologies, different fuel types, operation methods, combustion regimes, combustion phases, and aging conditions. The PB-ROS content and the chemical properties of the aerosols were quantified by a novel ROS analyzer using the DCFH (2′,7′-dichlorofluorescin) assay and a high-resolution time-of-flight aerosol mass spectrometer (HR-ToF-AMS). For all eight combustion devices tested, primary PB-ROS concentrations substantially increased upon aging. The level of primary and aged PB-ROS emission factors ($EF_{ROS}$) were dominated by the combustion device (within different combustion technologies) and to a greater extent by the combustion regimes: the variability within one device was much higher than the variability of $EF_{ROS}$ from different devices. Aged $EF_{ROS}$ under bad combustion conditions were $\sim 2$–80 times higher than under optimum combustion conditions. $EF_{ROS}$ from automatically operated combustion devices were on average 1 order of magnitude lower than those from manually operated devices, which indicates that automatic combustion devices operated at optimum conditions to achieve near-complete combustion should be employed to minimize PB-ROS emissions. The use of an electrostatic precipitator decreased the primary and aged ROS emissions by a factor of $\sim 1.5$ which is however still within the burn-to-burn variability. The parameters controlling the PB-ROS formation in secondary organic aerosol were investigated by employing a regression model, including the fractions of the mass-to-charge ratios $m/z$ 44 and 43 in secondary organic aerosol (SOA; $f_{44-\text{SOA}}$ and $f_{43-\text{SOA}}$), the OH exposure, and the total organic aerosol mass. The regression model results of the SC and PAM chamber aging experiments indicate that the PB-ROS content in SOA seems to increase with the SOA oxidation state, which initially increases with OH exposure and decreases with the additional partitioning of semi-volatile components with lower PB-ROS content at higher OA concentrations, while further aging seems to result in a decay of PB-ROS. The results and the special data analysis methods deployed in this study could provide a model for PB-ROS analysis of further wood or other combustion studies investigating different combustion conditions and aging methods.

# 1 Introduction

Numerous studies worldwide have shown a link between exposure to airborne particulate matter (PM) and morbidity and mortality (Beelen et al., 2013; Dockery et al., 1993; He et al., 2016), and a strong correlation of airborne PM with lung function (Lee et al., 2011; Pope et al., 2002; Adam et al., 2015; Hwang et al., 2015). The adverse health effects of PM are related to the aerosol chemical composition (Kelly and Fussell, 2012; Baltensperger et al., 2008). Residential wood combustion can contribute to 5–44 % of the total ambient $PM_{2.5}$ (particulate matter with a diameter smaller than 2.5 μm), depending on the environment (Zhang et al., 2010; Germain, 2005; USEPA, 2000; EEA, 2013; Ciarelli et al., 2017). In addition to PM, wood combustion emits a wide range of gaseous pollutants, including volatile organic compounds, upon which oxidation can form secondary organic aerosol (SOA). Although wood is considered to be a climate neutral source of energy, epidemiological studies suggest that wood smoke may contribute significantly to premature mortality (Boman et al., 2003; Johnston et al., 2012), because of its association with respiratory disease, cerebrovascular diseases and impaired lung function (Liu et al., 2017; Yap, 2008; Fullerton et al., 2011). Liu et al. (2017) found a 7.2 % increase in the risk of respiratory hospital admissions during days with high wildfire-specific $PM_{2.5}$ compared to non-wildfire smoke event days. Exposure to wood combustion particles may cause moderate inflammatory activity, cell death and DNA damage, and adverse effects to airway epithelia (Krapf et al., 2017; Tapanainen et al., 2012; Muala et al., 2015; Marabini et al., 2017). These adverse effects may be related to oxidative stress caused by free radicals induced by inhaled PM, which overwhelms the antioxidants in the body (Lobo et al., 2010).

This may happen via two pathways: (1) particles may contain reactive oxygen species (particle-bound reactive oxygen species, PB-ROS, exogenous), which act as oxidants in the biological system; (2) particles may contain transition metals or organic compounds like quinones, which generate reactive oxygen species by interaction with physiological species undergoing Fenton reactions and redox cycling. For the measurement of the latter property, several assays have been introduced, where the loss of reductants like dithiothreitol (DTT), glutathione (GSH) or ascorbic acid (AA) is measured (Cho et al., 2005; Verma et al., 2012, 2015; Charrier and Anastasio, 2012; Fang et al., 2016; Weber et al., 2018; Mudway et al., 2004; Li et al., 2003). Also cellular tests with an ROS probe have been developed to measure ROS induced by aerosols in a biological system (Landreman et al., 2008; Zhang et al., 2016; Tuet et al., 2017). The DCFH (2′,7′-dichlorofluorescin) assay has been shown to be sensitive to a broad range of organic peroxides, alkyl peroxide radicals, and hypochlorite, but not to components that are known to induce redox cycling (i.e., metal ions and quinones; Venkatachari and Hopke, 2008; Wang et al., 2011 TS3; King

and Weber, 2013; G. W. Fuller et al., 2014; S. J. Fuller et al., 2014; Zhou et al., 2018). The DCFH assay has fast response rates and a linear response to varying ROS concentrations, for which reason it was applied as a suitable measure for the PB-ROS concentration (Zhou et al., 1997; Venkatachari and Hopke, 2008; King and Weber, 2013, Zhou et al., 2018).

Several studies tried to establish links between such measurements and specific inflammatory biomarkers of oxidative stress in cell cultures or human subjects. For example Delfino et al. (2010, 2013) found that macrophage ROS generation from $PM_{2.5}$ was significantly positively associated with nitric oxide exhaled from elderly subjects and school children with persistent asthma. Janssen et al. (2015) reported a significant association between exhaled nitric oxide and increases in interleukin-6 in nasal lavage and the ROS generation measured by DTT and AA. Others found correlations between DTT activity and emergency department visits for asthma/wheezing and congestive heart failure (Bates et al., 2015; Fang et al., 2016). We are not aware of a study that relates measurements of ambient PB-ROS by DCFH with inflammatory biomarkers. However, it was shown that fresh SOA can release OH and $H_2O_2$ when dissolved in water (Wang et al., 2011 TS4; Tong et al. 2016). This phenomenon was attributed to labile peroxides contained in SOA, which were shown do decay with a rather short lifetime of less than 1 h (Krapf et al., 2016 TS5). Lakey et al. (2016) modeled the ROS produced in the human respiratory tract upon inhalation of PM and showed that the OH production rate from SOA can be as high as the $H_2O_2$ production rate from trace metals. This indicates that PB-ROS might also lead to oxidative stress. Obviously, further research is needed to establish a link between PB-ROS and health effects.

Based on these considerations we performed a study with a DCFH assay to focus on exogenous PB-ROS formed by wood combustion exhaust and during its atmospheric aging. For this purpose, a suite of smog chamber (SC) and potential aerosol mass (PAM) chamber experiments were conducted. As different types of wood, combustion devices, and combustion conditions result in varying levels of emissions (Johansson et al., 2004; Schmidl et al., 2011; Fitzpatrick et al., 2007; Heringa et al., 2011), eight wood combustion devices with variable combustion conditions were tested. Primary and aged biomass smoke generated under different combustion and aging conditions were characterized by an online ROS analyzer based on the DCFH assay coupled with an aerosol collector. Observations from this study provide more detailed evidence of the influence of combustion technology on the PB-ROS of the emitted PM compared to a previous similar study (Miljevic et al., 2010). We also show the variation of the PB-ROS content from primary and aged aerosols under different operation conditions. Further, the contribution of reactive oxygen species to aged organic aerosol generated with different aging tools was investigated to clarify the PB-ROS formation potential upon photo-oxidation. Re-

**Table 1.** Overview of combustion devices and test aspects.

| Aging conditions | Combustion devices | Test aspects |
| --- | --- | --- |
| PAM chamber ($T = \sim 38\,°C$, RH $= 20$–$25\,\%$) | pellet boiler (PB) | $EF_{ROS}$ of different burning regimes* ($\lambda^{++}$, $\lambda^{-}$, $\lambda^{opt}$) |
| | moving grate boiler (MGB) | $EF_{ROS}$ of full/part load; with/without electrostatic precipitator (ESP) |
| | pellet stove (PS) | $EF_{ROS}$ of different burning phases* |
| | log wood boiler (LWB) | $EF_{ROS}$ of different burning phases* |
| | log wood stove 1 (LWS1) | $EF_{ROS}$ of different burning phases* |
| | log wood stove 2 (LWS2) | $EF_{ROS}$ of different burning phases* |
| | log wood stove 3 (LWS3) | $EF_{ROS}$ of different burning phases* Secondary ROS formation |
| PSI-MSC ($\sim 7\,m^3$) $T = -10$ or $15\,°C$, RH $= 50\,\%$ | log wood stove 4 (LWS4) | flaming phase, aging temperature; secondary ROS formation |
| PSI-SSC ($\sim 27\,m^3$) Ambient $T$ ($22.5\,°C$), RH $= 50\,\%$ | log wood stove 4 (LWS4) | flaming phase; secondary ROS formation |

* The definitions of the burning regimes and burning phases are described in Sect. 2.2.

sults from these experiments may be directly compared with ambient measurements.

## 2 Experimental setup and methodology

We performed two sets of measurement campaigns, utilizing several wood combustion devices with different combustion conditions and two aging tools. First we present the different devices, then give a description of the PAM chamber and the Paul Scherrer Institute (PSI) mobile smog chamber (PSI-MSC, $\sim 7\,m^3$) and the PSI stationary smog chamber (PSI-SSC, $27\,m^3$) (Platt et al., 2013, 2014; Paulsen et al., 2005), including the experimental procedures, and finally discuss the combustion conditions and measurement strategy. An experiment schematic is shown in Fig. S1 in the Supplement. The combustion devices, experiment aging tools, and the test aspects are listed in Table 1.

### 2.1 Combustion devices

Eight combustion devices with different technologies were tested, including a pellet boiler (PB, automatic), a moving grate boiler equipped with electrostatic precipitator (MGB, automatic), a updraft combustion pellet stove (PS, automatic), a two-stage combustion downdraft log wood boiler (LWB, manual), two advanced two-stage combustion log wood stoves (LWS1, manual, updraft; LWS2, manual, updraft combustion when cold and downdraft combustion when hot), and two conventional single-stage combustion log wood stoves (LWS3, manual; LWS4, manual). In the following, we describe the different combustion devices.

– *PB:* Automatically operated pellet boiler, with two-stage updraft combustion and a nominal heat output of $15\,kW$, using wood pellets (EN certified, moisture content $7\,\%$) as the combustion fuel. Under optimum combustion conditions, the ideal air-to-fuel ratio ($\lambda$) is achieved leading to near-complete combustion and, consequently, the particle emissions are dominated by inorganic components which are contained in the pellets. The PB was also altered to enable the variation of the air-to-fuel ratio to investigate the influence of this parameter on the emissions. In this way, different combustion regimes could be achieved with this device; details are described in Sect. 2.2.

– *MGB:* Automatically operated industrial moving grate boiler with nominal heat output of $150\,kW$, operated with wood chips ($30\,\%$ moisture content). The grate has several zones where primary and secondary combustion air can be regulated.

– *PS, LWB, LWS1, LWS2, LWS3, and LWS4:* LWB, LWS1, and LWS2 are advanced stoves/boilers with two-stage combustion technology, while in LWS3 and LWS4 conventional single-stage updraft combustion is applied. PS is an automatically operated pellet stove with a nominal heat output of $6\,kW$ under full load. It possesses a ventilator for the injection of the combustion air. However, due to a relatively simple air control, the PS is operated at high $\lambda$. We also investigated part-load conditions at $3\,kW$. LWB, LWS1, LWS2, LWS3, and LWS4 are manually operated devices, with the nominal heat outputs of 30, 8, 4.6, 8, and $4.5\,kW$, respectively. Further, the LWS1 is equipped with a storage container for logs, which slide on the grate due to gravity. For all

four two-stage combustion devices (PS, LWB, LWS1, and LWS2) and one single-stage combustion device (LWS3), PB-ROS emissions from starting, flaming, and burn-out phases were investigated (details of the combustion phases are described in Sect. 2.3). In the case of the LWS4, only the flue gas from the flaming phase was injected into the smog chamber, where the $EF_{ROS}$ under different aging temperatures of $-10$ and $15\,°C$ were tested. In three of the log wood operated devices (LWS1, LWS2, and LWS3) dry (13–16 % moisture content) and wet logs (24–42 % moisture content) were investigated. In the PS, wheat pellets (manufactured from milling residues, moisture content 9 %) were tested in addition to conventional wood pellets (EN certified, moisture content of 7 %). In the LWS4, beech wood logs with a moisture content of $18 \pm 3$ % were used.

## 2.2 Combustion conditions

Two parameters are used to describe the combustion conditions, namely, the combustion regimes and the combustion phases. Combustion regimes are defined by the air fuel equivalence ratio ($\lambda$) (Nussbaumer and Kaltschmitt, 2000).

$$\lambda = \frac{O_{2,amb}\,[\%]}{O_{2,amb}\,[\%] - O_{2,flue\,gas}\,[\%]}, \tag{1}$$

where $O_{2,amb}$ and $O_{2,flue\,gas}$ are the oxygen contents in ambient air ($O_{2,amb} = 21$) and in the flue gas, respectively. Depending mainly on the level of excess air three combustion regimes are distinguished: lack of oxygen ($\lambda^-$), optimum combustion conditions ($\lambda^{opt}$), and (high) excess of oxygen ($\lambda^{++}$). Each of these is characterized by a different type of combustion particle, i.e., comprising mostly soot, salts, and condensable organic compounds, respectively (Nussbaumer and Lauber, 2010). It should be noted that in wood combustion $\lambda$ is always $> 1$. Consequently, $\lambda^-$ and $\lambda^{++}$ only describe $\lambda$ values which are clearly (for $\lambda^{++}$ at least 1.5-fold or higher) below or above $\lambda^{opt}$.

The three combustion regimes were achieved by changing the air-to-fuel ratio in the pellet boiler (PB). Optimum combustion conditions ($\lambda^{opt}$) were easily achieved by operating the PB under the designed optimum operation mode. High excess of oxygen ($\lambda^{++}$) compared to $\lambda^{opt}$ was obtained by additionally blowing air into the combustion chamber via the ignition tube. The lack of oxygen ($\lambda^-$) regime was obtained by manually closing the secondary combustion air inlet. It should be noted that in real life operation $\lambda^{++}$ and $\lambda^-$ conditions only occur with severe mal-operation. These conditions were investigated since they result in distinct emission characteristics (high non-methane volatile organic compound emissions during $\lambda^{++}$ and high soot emissions during $\lambda^-$ (Nussbaumer and Lauber, 2010).

In the MGB, part-load (50 kW) and full-load (150 kW) conditions, and the influence of an electrostatic precipitator (ESP) installed downstream of the combustion unit, were

tested. ESPs are widely used in both large- and small-scale wood combustion devices to reduce PM emissions (Bologa et al., 2011; Nussbaumer and Lauber, 2010).

Combustion phases in the log wood stoves, log wood boiler and pellet stove were classified using the modified combustion efficiency (MCE), defined as the molar ratio of the emitted $CO_2$ divided by CO plus $CO_2$ ($CO_2 / (CO + CO_2)$), in the flue gas after wood combustion (Ward and Radke, 1993). Each full combustion cycle includes three combustion phases: start phase (beginning of the burning cycle before MCE reaches 0.974), flaming phase (between start and burnout phase, with MCE > 0.974), and burnout phase (after flaming phase, with MCE < 0.974). As mentioned in Sect. 2.1, all three phases were obtained in the PS, LWB, LWS1, LWS2, and LWS3. In the PS, LWB, and LWS1, experiments started with a cold start, followed by a flaming phase and burn out. In the LWS2 and LWS3, after the first complete combustion cycle starting with a cold start, several full combustion cycles followed by adding new logs into the combustion chamber after the burn out was finished (warm start). In devices where the combustion phases were rapidly changing, the ROS analyzer was not able to separate these combustion phases due to a slow response time ($\sim 8$ min). Consequently, the single combustion phases, including the start, flaming, and burn out, and the combined combustion phases start + flaming or flaming + burn out were used for the PB-ROS analysis. In the LWS4, with which the experiments were conducted in the PSI-MSC (at temperatures of 263 and 288 K), and the PSI-SSC (at a temperature of 288 K), only emissions from the flaming phase were sampled.

## 2.3 Experimental procedures and aging tools

### 2.3.1 PAM chamber

Seven combustion devices (except LWS 4) were tested using the PAM chamber as an aging tool. The emissions were sampled through a heated line (473 K), diluted by a factor of $\sim 100$–150 using two ejector diluters in series (VKL 10, Palas GmbH), and then injected into the PAM chamber (see Fig. S1). The original concept of the PAM chamber is described by Kang et al. (2007). Briefly, the PAM chamber is a single $0.015\,m^3$ cylindrical glass chamber, flanked by two UV lamps. Prior to entering the PAM chamber, pure air ($1.6\,L\,min^{-1}$, humidified with a Nafion membrane, Perma Pure LLC) used as an OH precursor and a stream of diluted d9-butanol (98 %, Cambridge Isotope Laboratories) were merged with the incoming reactant flow. The OH exposure during aging was defined as the integral of the OH concentration over the reaction time, and was calculated from the decay of the d9-butanol, measured by a proton transfer reaction–mass spectrometer (PTR-MS 8000, Ionicon Analytik GmbH; Barmet et al., 2012). The total flow rate in the PAM chamber was maintained at $\sim 7\,L\,min^{-1}$, which

was the sum of the flow rates of the instruments and a supplementary flow, resulting in a residence time of approximately 2 min. The OH exposure was controlled by adjusting the UV light intensity to obtain different OH concentrations. An outer ring flow ($\sim 0.7 \, \mathrm{L \, min^{-1}}$), which was discarded, was used to minimize wall losses and the instrument sampled only from the inner flow of the PAM chamber ($\sim 6.3 \, \mathrm{L \, min^{-1}}$). The temperature in the PAM chamber was around 38 °C due to the lamps. Primary wood combustion emissions were characterized either before or after the PAM chamber when the lights were switched off. Aged emissions were characterized after the PAM chamber with lights on. All the experiments were conducted under OH exposures of $(1.1\text{–}2.0) \times 10^8 \, \mathrm{molec \, cm^{-3} \, h}$ CE2 which corresponds to $\sim 4.5\text{–}8$ days of aging in ambient by assuming a mean daily OH concentration of $1 \times 10^6 \, \mathrm{molec \, cm^{-3}}$. The applicability of the PAM chamber to measure wood combustion emissions has been shown in a previous study (Bruns et al., 2015).

### 2.3.2 Smog chamber aging

The second set of experiments was conducted in the PSI mobile smog chamber (PSI-MSC, $\sim 7 \, \mathrm{m^3}$) at temperatures of 263 and 288 K, and the PSI stationary smog chamber (PSI-SSC, $27 \, \mathrm{m^3}$) at 295.5 K. An overview of the experimental setup is also shown in Fig. S1. In general, three pieces of dry beech logs, four pieces of kindling, and three fire starters were combusted in LWS4 for average ($2.9 \pm 0.3$ kg) experiments and nine pieces dry beech logs, eight pieces kindling, and four fire starters were combusted for high (5.1 kg) load experiments (details in Sect. 2.1). The wood moisture content was $19 \pm 2\%$. Only emissions during the flaming phase with a modified combustion efficiency (MCEs) in the range from 0.974 to 0.978 were sampled. Emissions were sampled for 11–21 min and injected into the PSI-MSC using an ejection diluter, yielding a total dilution factor of 100 to 200. Hydroxyl radical (OH) concentrations in the chamber are controlled by continuous injection of nitrous acid into the smog chamber (after the characterization of the primary emissions as described below in Sect. 3.1), which produces OH upon irradiation by UV lights (Platt et al., 2013). The OH exposure was estimated by monitoring the decay of d9-butanol (butanol-D9, 98 %, Cambridge Isotope Laboratories) following a single injection before the UV lights were turned on. In all five experiments conducted in the PSI-MSC, the aging time lasted 4.5–6 h. The OH exposure was $2.6\text{–}4.8 \times 10^7 \, \mathrm{molec \, cm^{-3} \, h}$, which corresponds to $\sim 1\text{–}2$ days of aging in ambient by assuming a mean daily OH concentration of $1 \times 10^6 \, \mathrm{molec \, cm^{-3}}$. More details about some of the PSI-MSC experiments of this campaign can also be found in Bruns et al. (2016, 2017). One additional experiment was conducted in the PSI-SSC, with an OH exposure up to $4.0 \times 10^8 \, \mathrm{molec \, cm^{-3} \, h}$ TS6, equivalent to $\sim 17$ days of aging assuming a mean daily OH concentration

of $1 \times 10^6 \, \mathrm{molec \, cm^{-3}}$, extending the aging range beyond the range achieved by the PAM chamber ($\sim 1\text{–}8.5$ days).

### 2.4 Particle-phase characterization

The non-refractory particle chemical composition was measured using a high-resolution time-of-flight aerosol mass spectrometer (HR-ToF-AMS; flow rate: $0.1 \, \mathrm{L \, min^{-1}}$, Aerodyne Research Inc.; DeCarlo et al., 2006). The HR-ToF-AMS measured the total organic aerosol (OA), $SO_4^{2-}$, $NO_3^-$, $NH_4^+$, $Cl^-$, and the two most dominant oxygen-containing ions in the OA spectra, i.e., the mass-to-charge ratios $m/z$ 44 (Org44, mostly $CO_2^+$) and $m/z$ 43 (Org43, mainly $C_2H_3O^+$ for the oxygenated OA and $C_3H_7^+$ for the hydrocarbon-like OA; Ng et al., 2011). Equivalent black carbon (eBC) was determined using an Aethalometer (AE33, Magee Scientific; flow rate: $2 \, \mathrm{L \, min^{-1}}$, Drinovec et al., 2015).

The particle-bound ROS was characterized by an online ROS analyzer (flow rate: $1.7 \, \mathrm{L \, min^{-1}}$) (Zhou et al., 2018). The aerosols particles were collected in a mist chamber-type aerosol collector, dissolved into water, and mixed with a 2′,7′-dichlorofluorescin (DCFH)/horseradish peroxidase solution. The ROS converts DCFH to DCF, which is detected by fluorescence and quantified as nM-$H_2O_2$ equivalents. The time resolution of the online ROS analyzer was $\sim 8$ min, preventing resolving brief discrete combustion phases. Therefore, different methods were used to calculate the average PB-ROS emissions under different conditions:

1. average (Fig. S2a): utilized when the combustion conditions were relatively stable and sufficiently long to yield a stable ROS signal;

2. integrated average (Fig. S2b): in cases of variable combustion conditions, the ROS signal was integrated over the measurement period which could include one or several phases from the same burn;

3. extrapolation + integrated average (Fig. S2: panels 2c_1 and c_2): when the combustion conditions were variable and the background could not be measured between two combustion conditions due to the time resolution of the ROS instrument. We extrapolate each measurement to the background value and then make the integrated average calculation as described above.

The various definitions for PB-ROS and related aerosol characteristics are presented below:

– *PB-ROS emission factors* ($EF_{ROS}$). PB-ROS emission factors ($EF_{ROS}$) were calculated as the amount of PB-ROS in nmol-$H_2O_2$ equivalents per kilogram wood burnt, using Eq. (2):

$$EF_{ROS} = \frac{n_{ROS}}{M_C} C_{wood} \cong$$

$$\frac{[n_{ROS}]}{\sum([\rho C_{CO_2}] + [\rho C_{CO}] + [\rho C_{CH_4}] + [\rho C_{VOC}] + [\rho C_{eBC}] + [\rho C_{OC}])} C_{wood}, \quad (2)$$

where $[n_{ROS}]$ is the background-corrected concentration of PB-ROS (nmol m$^{-3}$) in the emitted particles either before (primary PB-ROS) or after aging (aged PB-ROS), $[\rho C_x]$ are the carbon mass concentrations calculated from the background-corrected, carbon-containing species where $x$ includes $CO_2$, CO, $CH_4$, volatile organic compounds (VOC), eBC, and particulate organic carbon (OC). $M_C$ is the carbon mass burnt and $C_{wood}$ represents the average carbon fraction of the wood fuel, $\sim 0.46$, measured in this study using an elemental analyzer. OC data were obtained from AMS measurements. Similarly, the organic aerosol (OA) emission factors (EF$_{OA}$) were calculated by replacing the PB-ROS concentration by OA.

– *PB-ROS fraction*. In order to study the PB-ROS formation during aging, the secondary PB-ROS fraction ($f_{ROS-SOA}$) is introduced. It expresses the amount of secondary PB-ROS (ROS$_S$ = aged ROS − primary ROS) per amount of secondary organic aerosol (SOA) formed during aging and as calculated from Eq. (2)

$$f_{ROS-SOA} = \frac{ROS_S}{SOA} \qquad (3)$$

Secondary organic aerosol (SOA) and secondary PB-ROS (ROS$_S$) were calculated by subtracting primary organic aerosol (POA) and primary PB-ROS (ROS$_P$) from the total OA and aged PB-ROS, respectively, assuming ROS$_S$ and POA to only be lost to the chamber wall at the same rate as eBC but otherwise to remain constant during aging. Although both quantities may not be conserved, a decrease of both does partially compensate CE3 in the PB-ROS fraction. In the SC experiments, POA is defined as the OA mass before lights on, while SOA is estimated as the difference between total OA and the time-dependent POA mass accounting for particle wall loss. Wall loss rates for POA and SOA were assumed to be the same as that of the measured eBC. In PAM aging experiments, each experiment had a certain POA (measurements before PAM or after PAM with lights of) and SOA (measurements after PAM with lights on).

– $f_{44-SOA}$ *and* $f_{43-SOA}$. To express the degree of oxygenation of SOA, the fraction of secondary Org44 and Org43 in SOA (represented as $f_{44-SOA}$ and $f_{43-SOA}$) is introduced, which is calculated from Eq. (4)

$$f_{44-SOA} = \frac{Org_{44-SOA}}{SOA}; \quad f_{43-SOA} = \frac{Org_{43-SOA}}{SOA}, \quad (4)$$

where Org$_{44-SOA}$ is the difference of total Org44 and primary Org44, Org$_{43-SOA}$ is the difference of total Org43 and primary Org43 and using the same procedure as for the SOA calculation mentioned above.

– *Wall loss correction*. The wall loss correction in the SC was done by assuming the same losses for all particle components as for the inert tracer eBC. The wall loss-corrected concentration of OA or PB-ROS ($X$) can be derived using Eq. (5):

$$X_{WLC}(t) = X_{meas}(t) \times \frac{BC(t0)}{BC(t)}, \qquad (5)$$

where $X_{meas}(t)$ refers to the concentration of $X$ measured at time $t$. BC ($t_0$) and BC ($t$) are the concentrations of BC when lights were switched on and at time $t$, respectively.

## 2.5 Gas-phase characterization

During the PAM chamber experiments, total volatile organic compounds (VOC) and $CH_4$ (using a flame ionization detector (FID) with a non-methane cutter, model 109A, J.U.M Engineering), CO and NO (with a non-dispersive infrared analyzer, Ultramat 23 Siemens), and $O_2$ (using a paramagnetic oxygen analyzer, Ultramat 23 Siemens) were determined in the hot undiluted flue gas. In SC aging experiments CO was measured with a cavity ring-down spectrometer (G2401, Picarro, Inc.). In all experiments, the composition of VOCs was determined by a the PTR-MS 8000 (Ionicon Analytik GmbH). For $CO_2$ a cavity ring-down spectrometer (G2401, Picarro, Inc.) was used in the SC aging experiments and a non-dispersive infrared (NDIR) analyzer (model LI-820, LI-COR$^{®}$) in the PAM chamber aging experiments.

## 3 Results and discussion

### 3.1 Primary and aged PB-ROS emission factors (EF$_{ROS}$)

The PB-ROS and OA emission factors are presented in Table 2 for all combustion conditions, together with the number of tests, the combustion efficiency (MCE), the air-to-fuel ratio ($\lambda$), and the aerosol bulk properties determined with the AMS (OM : OC, O : C and H : C ratios). The given values are the 25th and 75th percentiles of averages from several experiments, and the data points considered for the calculations were restricted to the time period of the PB-ROS measurements. As shown in Fig. 1, PB-ROS emission factors (EF$_{ROS}$) for primary and aged OA were highly variable depending on the combustion conditions and devices. For all devices and combustion conditions, a substantial enhancement in the EF$_{ROS}$ is observed with aging, indicating the importance of secondary PB-ROS production. The PB-ROS enhancement factor, defined as the ratio between aged and primary EF$_{ROS}$, range between 4 and 20, with lower values for MGB ($\sim 4$) and PB under $\lambda^{opt}$ combustion conditions ($\sim 6$),

**Table 2.** Characterization of primary emissions from PAM chamber and SC aging experiments*.

| Devices | Test aspects | | No. of tests | MCE | $\lambda$ (nmol kg$^{-1}$) | PB-ROS (mg kg$^{-1}$) | Total PM (mg kg$^{-1}$) | Org | OM : OC | O : C | H : C |
|---|---|---|---|---|---|---|---|---|---|---|---|
| PB | $\lambda^-$ | | 3 | [0.991, 0.992] | [1.29, 1.30] | [345, 882] | [246, 301] | [56, 62] | [2.1, 2.4] | [0.7, 0.9] | [1.3, 1.4] |
| | $\lambda^{opt}$ | | 7 | [0.999, 0.999] | [1.59, 1.64] | [288, 2325] | [50, 69] | [22, 29] | [2.7, 2.8] | [1.1, 1.2] | [1.3, 1.4] |
| | $\lambda^{++}$ | | 15 | [0.963, 0.983] | [3.02, 3.11] | [1940, 5944] | [33, 61] | [15, 26] | [2.5, 2.6] | [0.9, 1.0] | [0.9, 1.0] |
| MGB | Full load | Before ESP | 5 | [0.999, 0.999] | [1.99, 2.04] | [1758, 2034] | [65, 100] | [27, 48] | [3.1, 3.1] | [1.4, 1.4] | [1.1, 1.3] |
| | | After ESP | 3 | [0.999, 0.999] | [3.91, 3.99] | [775, 1098] | [3, 4] | [1, 2] | [2.3, 2.7] | [0.7, 1.0] | [1.2, 1.4] |
| | Part load | Before ESP | 6 | [0.999, 0.999] | [2.12, 2.30] | [780, 4083] | [19, 25] | [8, 9] | [2.1, 2.3] | [0.6, 0.8] | [1.1, 1.3] |
| PS | All burning phases | | 5 | [0.989, 0.995] | [4.97, 7.59] | [5376, 36 415] | [204, 625] | [60, 427] | [2.2, 2.5] | [0.8, 1.0] | [1.1, 1.3] |
| LWB | All burning phases | | 20 | [0.904, 0.999] | [1.47, 2.49] | [4307, 27 590] | [262, 741] | [111, 277] | [2.5, 2.9] | [1.0, 1.4] | [1.1, 1.2] |
| LWS1 | All burning phases | | 6 | [0.850, 0.933] | [3.57, 7.05] | [5915, 52 528] | [381, 572] | [142, 2.4] | [2.3, 1.0] | [0.9, 1.2] | [1.2, |
| LWS2 | All burning phases | | 6 | [0.948, 0.976] | [3.51, 4.31] | [141 457, 249 755] | [49, 98] | [49, 98] | [2.3, 2.4] | [0.9, 1.0] | [1.2, 1.3] |
| LWS3 | All burning phases | | 19 | [0.930, 0.968] | [4.61, 9.57] | [12 160, 61 258] | [151, 356] | [14, 55] | [1.9, 2.1] | [0.5, 0.6] | [1.4, 1.6] |
| LWS4 | Flaming | | 5 | [0.972, 0.975] | [3.0, 3.6] | [37 766, 57 403] | [171, 440] | [83, 162] | [1.6, 1.7] | [0.30, 0.45] | [1.3, 1.5] |

* Values of each parameter are described as [a, b], where a and b represent the 25th and 75th percentiles of the averages from several experiments, and the data points considered for the calculations were restricted to the time period of the PB-ROS measurements.

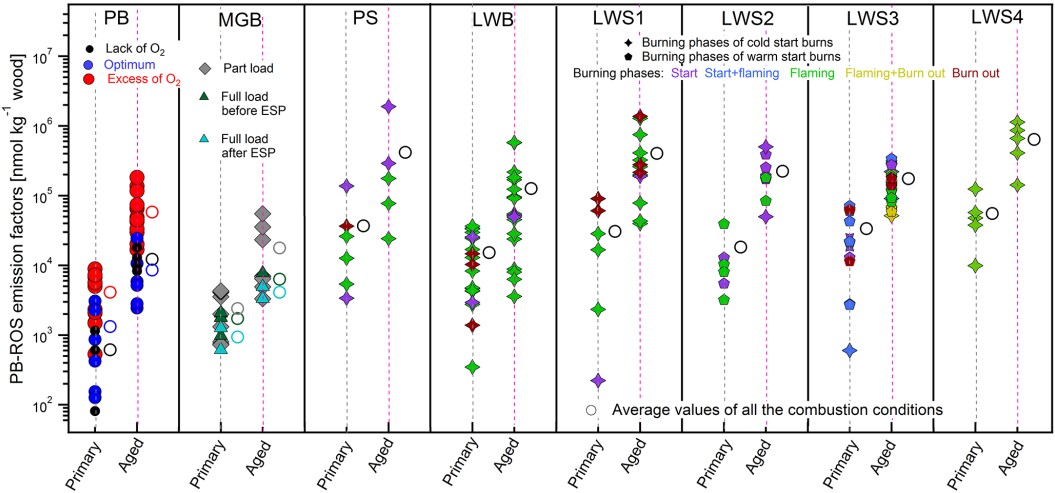

**Figure 1.** PB-ROS emission factors (EF$_{ROS}$) for all tested combustion devices under different operating and aging conditions. Open circles symbols represent the average values of all the experimental data points for each condition. PB denotes pellet boiler; MGB, moving grate boiler; PS, pellet stove; LWB, log wood boiler; LWS$n$, log wood stove $n$ ($n$ = 1, 2, 3, 4). Each data point represents one experiment. For each device, primary EF$_{ROS}$ appear on the left side (gray dashed line) and aged EF$_{ROS}$ on the right side (pink dashed line).

and higher values for PB under $\lambda^-$ and $\lambda^{++}$ combustion conditions ($> 10$). The PB-ROS enhancement factors for all log wood stoves as well as LWB are comparable, with an average value around 10.

The variability in the $EF_{ROS}$ in primary and aged OA for one device is much higher than the variability between average emission factors for different devices, spanning almost 2 orders of magnitudes. Despite this, $EF_{ROS}$ from PB and MGB (80–8890 nmol kg$^{-1}$ wood and $2440$–$1.83 \times 10^5$ nmol kg$^{-1}$ wood for primary and aged emissions, respectively) are on average 1 order of magnitude lower than those from PS, LWB, and LWS1–4 ($220$–$1.89 \times 10^6$ nmol kg$^{-1}$ wood and $3570$–$1.1 \times 10^6$ nmol kg$^{-1}$ wood for primary and aged emissions, respectively). These results clearly indicate differences due to the combustion technology, as a general rule, $EF_{ROS}$ were lowest for automatically operated devices and higher for manually operated devices: PB and MGB are automatically operated and the primary and secondary air supply as well as the fuel feeding is controlled permanently, while LWB and LWS1–4 are manually operated. The PS is automatically operated but is operated at high $\lambda$ and exhibits similar $EF_{ROS}$ to the manual devices. Part of the $EF_{ROS}$ variability within each device can be ascribed to the combustion phase, with higher emission factors for the starting and burn-out phases compared to the flaming/stable phase. This is especially true for the aged emissions from the PS ($EF_{ROS}$ of the start phases was on average 13 times higher than the flaming phase; Mann–Whitney, $p$ value $= 0.06$), the LWS2 ($EF_{ROS}$ of the start phases was on average 1.7 times higher than the flaming phase, Mann–Whitney, $p$ value $= 0.24$, not significant), and the LWS3 ($EF_{ROS}$ of the start and burn-out phases were on average 1.5 times higher than the flaming and flaming + burn-out phase; Mann–Whitney, $p$ value $= 0.07$).

For the automatically operated MGB, the primary $EF_{ROS}$ did not statistically differ between part- and full-load operation (Mann–Whitney, $p$ value $= 0.95$). However, the aged $EF_{ROS}$ was a factor of $\sim 3$ higher for part load than for full load (Mann–Whitney, $p$ value $= 0.23$). The use of the electrostatic precipitator decreased primary and aged ROS emissions, on average by a factor of $\sim 1.5$ times, however, these differences are not statistically significant (Mann–Whitney, $p$ value $= 0.12$ for both primary and aged emissions) and are within the burn-to-burn variability.

For PB, the combustion operation could be systematically varied to investigate the influence of air-to-fuel ratio on PB-ROS and OA emission factors before and after aging. The $EF_{ROS}$ were highest under $\lambda^{++}$ conditions for both primary and aged emissions, with average values of 4100 and $5.8 \times 10^4$ nmol kg$^{-1}$ wood burnt, respectively (Fig. 1 and Table 2). Primary PB-ROS emissions under $\lambda^{opt}$ conditions did not statistically differ from $\lambda^-$ conditions (Mann—Whitney, $p$ value $= 0.43$), but on average 7 and 3 times lower than that obtained under $\lambda^{++}$ conditions, respectively (Mann—Whitney, $p$ value $< 0.005$ for both cases). The aged $EF_{ROS}$ under $\lambda^{opt}$ and $\lambda^-$ were also quite similar (Mann—Whitney, $p$ value $= 0.20$), but with average values 8 and 5.5 times lower than obtained under $\lambda^{++}$ conditions, respectively (Mann–Whitney, $p$ value $= 0.02$ for both cases). This shows that the air-to-fuel ratio has a significant effect on the PB-ROS emissions, which will be investigated for all devices hereafter.

## 3.2    Aged $EF_{ROS}$ under different combustion regimes

Figure 2 shows the aged $EF_{ROS}$ of the eight devices as a function of $\lambda$. Similar to PB, as already described above, a clear increase of $EF_{ROS}$ in the aged aerosol can be observed with increasing $\lambda$ values, with $\sim 2$–80 times higher aged $EF_{ROS}$ values under bad combustion conditions than under optimum combustion conditions, although the extent of the increase and the overall trend were not the same for all individual devices. In the MGB all the burns occurred at $2.0 < \lambda < 2.2$, leading to aged $EF_{ROS}$ (without ESP) in line with those from the PB between $\lambda^{opt}$ ($\lambda = 1.6$) and $\lambda^{++}$ ($\lambda$ ranged from 2.7 to 3.4). The combustion in all stoves (PS, LWS1–4) exhibited higher $\lambda$ ($\lambda > 2.2$) due to a less controlled air supply leading to less efficient combustion. In this excess of oxygen range, aged $EF_{ROS}$ ranged between $1.68 \times 10^4$ nmol kg$^{-1}$ wood and $1.38 \times 10^6$ nmol kg$^{-1}$ for $\lambda$ values between 2.2 and 17.6, where all aged $EF_{ROS}$ were high but without any systematic trend with $\lambda$, suggesting that other parameters may influence PB-ROS emissions as well. The LWB follows a different trend, where the aged $EF_{ROS}$ increase sharply with $\lambda$, starting at lower $\lambda$ values than the other manually operated devices. Aged $EF_{ROS}$ for LWB ranged from 3530 to $5.79 \times 10^5$ nmol kg$^{-1}$ wood within the $\lambda$-range of 1.5–2.6. Although trends in Fig. 2 show differences between devices, they highlight quite readily the important influence of the combustion conditions on aged $EF_{ROS}$.

While the combustion efficiency was found to have a strong influence on aged $EF_{ROS}$, the latter varies considerably, by a factor of 3–50, within the same combustion regime but for different combustion devices. In Fig. 3, we investigate to which extent this variability in aged $EF_{ROS}$ is related to the variability in the bulk OA emissions. The high correlation (Pearson's $R = 0.92$) observed in Fig. 3 suggests that changes in aged $EF_{OA}$ explain a great fraction of the variability in aged $EF_{ROS}$, implying that this variation is inherent to wood combustion conditions. Nonetheless, additional unexplained variation was observed between the two variables in Fig. 3, with the aged PB-ROS emission factors varying by a factor of 2.6 on average for the same aged $EF_{OA}$. To elucidate the reasons behind this variability, we investigate in the following the parameters controlling the secondary PB-ROS formation and its content in OA upon aging.

## 3.3    Influence of aging conditions on PB-ROS formation

– *Regression model setup and performance.* In this section, we seek to evaluate the relationship between the

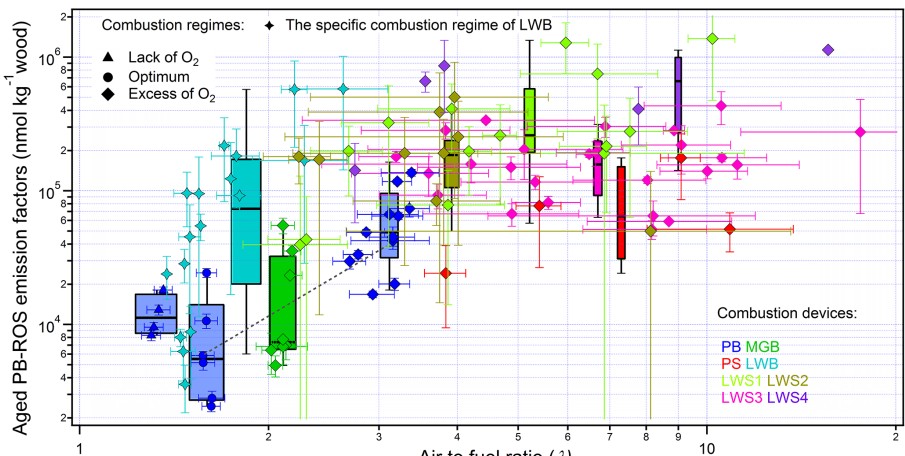

**Figure 2.** Aged PB-ROS emission factors (EF$_{ROS}$) from different combustion regimes and combustion devices. The grey dashed line represents the EF$_{ROS}$ increase with λ for the PB. The error bars of the $y$ axis of the data points denote the propagation of the uncertainty ($\delta = \sqrt{\delta_1^2 + \delta_2^2}$, with $\delta_1$ and $\delta_2$ representing the standard deviation of the averaged aged PB-ROS and aged OA of the measurement time periods, respectively.); the error bars of the $x$ axis of the data points denote the standard deviation of the averaged λ of the measurement time periods.

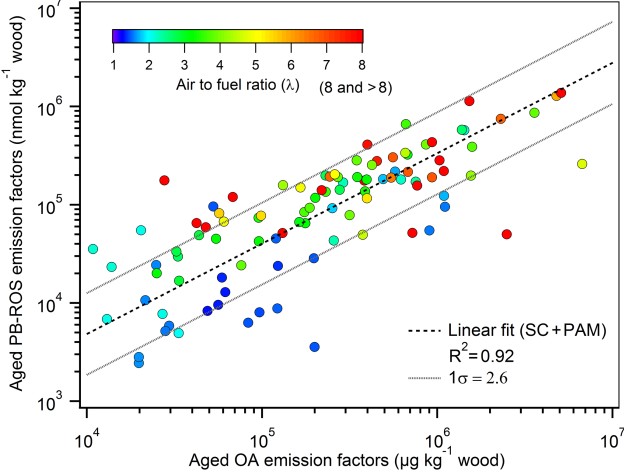

**Figure 3.** Aged PB-ROS emission factors vs. aged OA emission factors. Marker color corresponds to the air-to-fuel ratio (λ). Fitting equation, $\log_{10}(\text{EF}_{ROS}) = 0.92\log_{10}(\text{EF}_{OA})$, indicates that the relationship between aged PB-ROS and aged OA is almost linear. The geometric standard deviation obtained from the fit is 2.6, suggesting that the aged PB-ROS content of aged OA may vary significantly depending on the combustion and atmospheric aging conditions.

fraction of PB-ROS in SOA, $f_{ROS-SOA}$, and parameters controlling its formation. To exclude the influence of the combustion devices, the data obtained using the LWS4 in the SC experiments and using the LWS3 in the PAM chamber experiments was chosen for the analysis, as the LWS3 and LWS4 are both conventional single-stage combustion devices. Four different parameters were investigated, including $f_{43-SOA}$ and $f_{44-SOA}$, the OH ex-

posure, and the organic aerosol mass, by running the regression model as follows:

$$f_{ROS-SOA} = f_{44-SOA} + c \times f_{43-SOA}a \times \text{SOA} \quad (6)$$
$$+ b \times + d \times (\text{OH exposure}) + \text{intercept},$$

where $f_{43-SOA}$ and $f_{44-SOA}$ are supposed to represent the contributions of moderately oxygenated components (e.g., alcohols and carbonyls) and highly oxygenated components (e.g., carboxylic acids and peroxides), respectively. The organic aerosol mass may influence the fraction of PB-ROS in SOA, by affecting the amount of condensing semi-volatile species, which might be characterized by different $f_{ROS-SOA}$ compared to low-volatility species dominating at low organic aerosol mass. The aim of the multiple regression analysis used here is to extract the influence of different aging factors on the observed variance in $f_{ROS-SOA}$ (the 2.6 factor variance described in Fig. 3), and to assess the magnitude of their influence. We do not, however, propose using the model and the model coefficients for a deterministic explanation of PB-ROS formation.

Since the dependent variable, $f_{ROS-SOA}$, and the predictors considered are log-normally distributed – typical of concentrations and contributions – we have log-transformed the data before the multiple regression analysis. We note though that this step did not influence the conclusions of the analysis, as a multi-linear model applied to the raw data without a prior log-transformation suggests a similar relationship between $f_{ROS-SOA}$ and the predictors. Both models reasonably represented the measurements (∼20 % error, Fig. S4),

but log-transforming the data allowed for a better capturing of lower $f_{ROS-SOA}$ and a less skewed distribution of the model residuals (Fig. S4). We did not consider any interactions between the different regressors, as this is taken into account through the prior log-transformation of the data. For the parameterization, we only considered the SC data and will discuss whether the PAM chamber data could be satisfactorily explained by the same parameterization or whether the amount of PB-ROS formed under different conditions, with high OH concentrations in the PAM chamber, is different.

We note that the different predictors exhibit some degree of collinearity. For example, not unexpectedly, $f_{44-SOA}$ significantly increases with aging ($R^2$ between $f_{44-SOA}$ and OH exposure $= 0.68$), while $f_{43-SOA}$ increases with the amount of organic aerosol in the smog chamber ($R^2 = 0.56$), possibly due to the enhanced partitioning of the moderately oxidized organic species at higher absorptive mass (Pfaffenberger et al., 2013). Both variables, $f_{44-SOA}$ and $f_{43-SOA}$, are slightly inversely correlated ($R^2 = 0.26$). Therefore, prior to the regression analysis we inspected the severity of multicollinearity by computing the variance inflation factors (VIF) for all four predictors. All VIF values were between 2.5 and 6 (highest for $f_{44}$-SOA and for OH exposure), indicating a moderate degree of multicollinearity (VIF values above 10 would be related to excessive multicollinearity). While a direct consequence of multicollinearity is an increased probability of erroneously rejecting the dependence of $f_{ROS-SOA}$ on one of the factors, a type two error, the regression analysis suggests that the dependence of $f_{ROS-SOA}$ on all parameters is significant ($p < 10^{-6}$).

– *Model results for SC data.* The correlation between $f_{ROS-SOA}$ and the most important regressors is shown in Fig. 4. The analysis suggests that the greatest share of explained variability in $f_{ROS-SOA}$ could be attributed to $f_{44-SOA}$. An increase in $f_{44-SOA}$ by 1 geometric standard deviation (a factor of 1.45) resulted in our case in a doubling of the secondary PB-ROS fraction ($f_{ROS-SOA}$). This indicates that more oxygenated compounds are preferentially PB-ROS active compared to others.

The second most important parameter controlling the secondary aerosol PB-ROS content under our conditions is found to be the OH exposure. An increase in OH exposure by 1 geometric standard deviation (a factor of 2.7) resulted in our case in a 60 % decrease of the PB-ROS fraction in SOA ($f_{ROS-SOA}$). We note that the considerable effect size of this variable stems from its large variability, spanning a dynamic range of 2.5 orders of magnitude (e.g., $\sim 4$ times more variation in OH exposure compared to $f_{44-SOA}$ would be required to achieve

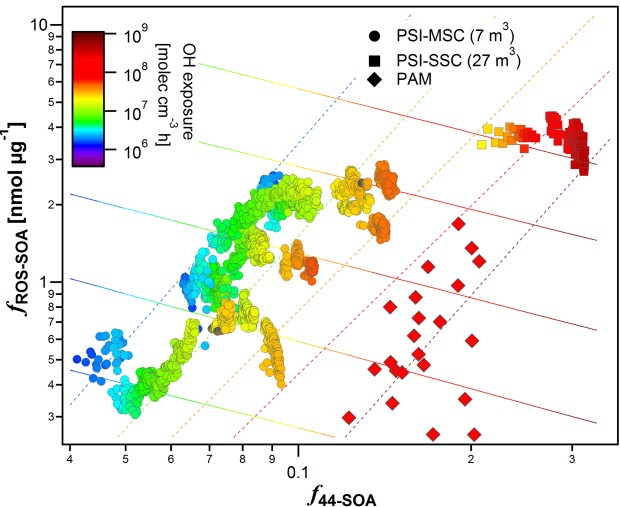

**Figure 4.** Variation of the fraction of PB-ROS in SOA, $f_{ROS-SOA}$, with the fraction of $m/z$ 44 in the total signal SOA as measured by the AMS ($f_{44-SOA}$) color coded with the OH exposure estimated from the decay of d9-butanol measured by the PTR-ToF-MS. Data are collected from two different smog chambers (SC) and from the PAM chamber. Dashed lines are isopleths of constant OH exposures, while solid lines are obtained by isolating the effect of OH exposure from other variables. To help discerning different experiments performed in SC, the same content in this figure is plotted again in Fig. S3, where those SC experiments are labeled by different numbers.

the same effect on $f_{ROS-SOA}$). The anti-correlation between OH exposure and $f_{ROS-SOA}$ indicates that the initially formed PB-ROS are prone to further reactions, consistent with previous observations of rapid peroxide (Krapf et al., 2016) and PB-ROS (Zhou et al., 2018) decay. The mechanism by which PB-ROS evolves remains uncertain, but may involve the oxidation of PB-ROS-related molecules by OH as well as their photolysis and unimolecular decay reaction. We note that the OH exposure increases the oxidation state of the aerosol, here represented by $f_{44-SOA}$, thereby indirectly increasing the PB-ROS content, especially in the beginning of the experiment. Therefore, the actual effect of OH exposure on $f_{ROS-SOA}$ could only be revealed when it was isolated from the $f_{44-SOA}$ effect (see Fig. 4).

The analysis suggests that $f_{43-SOA}$ and the organic mass concentrations exhibit a low, but statistically significant, effect on $f_{ROS-SOA}$ (Fig. S5). Their increase results in a decrease in the secondary PB-ROS content, consistent with the increased partitioning of moderately oxygenated components, which seem to contain less PB-ROS.

– *Comparison between SC and PAM chamber data.* The conditions in the PAM chamber are different from those

in the SC. PAM chamber experiments were conducted at high OH exposures of $\sim 10^8$ molecules cm$^{-3}$ h, where the resulting aerosol was highly oxygenated. However, the secondary PB-ROS content of the aerosol in the PAM chamber was largely within the expected range, following consistent trends with high OH exposures and high $f_{44-\text{SOA}}$ as in the SC (Fig. 4). We examined in more detail whether the regression model parameters obtained from the SC could faithfully represent the $f_{\text{ROS}-\text{SOA}}$ measured in the PAM chamber. Indeed, the model was capable of predicting, within uncertainties ($2\sigma$), the $f_{\text{ROS}-\text{SOA}}$ measured in the PAM chamber for low organic aerosol concentrations (average 21 µg m$^{-3}$), but considerably (factor of 3 on average) overestimated $f_{\text{ROS}-\text{SOA}}$ at higher concentrations (average 68 µg m$^{-3}$). This is because such a range of concentrations at high OH exposures and high $f_{44-\text{SOA}}$ was not included in the training dataset, and as a result the model slightly underestimated the effect of OA concentration on $f_{\text{ROS}-\text{SOA}}$ (e.g., a three-fold increase in OA concentration in the PAM chamber results in a decrease of $f_{\text{ROS}-\text{SOA}}$ by 45 %, while the model suggests that the same increase would only result in a 10 % decrease). Despite this, for similar conditions $f_{\text{ROS}-\text{SOA}}$ measured in the PAM chamber and the SC were similar within our uncertainties. We also note that this slight bias does not affect the main conclusions of the analysis: the secondary PB-ROS content seems to initially increase with the SOA oxidation state, which increases with OH exposure and decreases with the additional partitioning of semi-volatile components with lower secondary PB-ROS content at higher SOA concentrations, while further aging seems to result in a decay of PB-ROS.

# 4   Summary and conclusions

In this study, eight wood combustion devices for log wood, pellets, and wood chips, denoted as log wood boiler (LWB), log wood stove 1 (LWS1), log wood stove 2 (LWS2), log wood stove 3 (LWS3), log wood stove 4 (LWS4), pellet boiler (PB), pellet stove (PS), and moving grate boiler (MGB), were tested. Experiments were conducted in a suite of aging tools, including the Paul Scherrer Institute mobile smog chamber (PSI-MSC, $\sim 7$ m$^3$, OH exposure: $(2.6-4.8) \times 10^7$ molec cm$^{-3}$ h), the Paul Scherrer Institute stationary smog chamber (PSI-SSC, 27 m$^3$, OH exposure: $(0.13-40) \times 10^7$ molec cm$^{-3}$ h), and the potential aerosol mass chamber (PAM chamber, OH exposure: $(11-20) \times 10^7$ molec cm$^{-3}$ h), to investigate the particle-bound reactive oxygen species (PB-ROS) formation potential of primary and aged wood combustion emissions from different combustion devices and conditions. The influence of combustion technologies, wood types (wood logs, wood pellets,

and wood chips), operation type (e.g., with/without ESP, automatic vs. manual operation), combustion regime (different air-to-fuel ratio ($\lambda$) ranging from low ($\lambda^-$), to optimum ($\lambda^{\text{opt}}$), to high values ($\lambda^{++}$)), combustion phases (start, flaming, burn out), and aging conditions (SC aging/PAM chamber aging) on PB-ROS emission factors (EF$_{\text{ROS}}$) were investigated. Results show that EF$_{\text{ROS}}$ for primary and aged OA were highly variable depending on the combustion conditions and devices. For all devices and combustion conditions, EF$_{\text{ROS}}$ substantially increased upon aging, indicating the secondary production of PB-ROS. The PB-ROS enhancement factors ranged between 4 and 20, with lower values for the MGB ($\sim 4$) and PB under $\lambda^{\text{opt}}$ combustion conditions ($\sim 6$), and higher values for the PB under $\lambda^-$ and $\lambda^{++}$ combustion conditions ($> 10$). The PB-ROS enhancement factors for all log wood stoves and the LWB were comparable, with an average value around 10.

The variability in the EF$_{\text{ROS}}$ in primary and aged OA for a single device was much higher than the variability between emission factors from different devices. A part of this variability within each device could be ascribed to the combustion phase, with higher emission factors for the starting and burn-out phases compared to the flaming phase. This was especially true for the aged emissions from the PS, LWS2, and LWS3. Despite this, EF$_{\text{ROS}}$ values from the PB and MGB were on average 1 order of magnitude lower than those from the PS, LWB, and LWS1–4. This indicates that applying automatic combustion devices operated at optimum conditions, to achieve near-complete combustion, is most effective at minimizing PB-ROS, in addition to those of POA, SOA, and BC. Although the EF$_{\text{ROS}}$ showed somewhat different trends between devices with varying $\lambda$, a clear increase of EF$_{\text{ROS}}$ in the aged aerosol can be observed from optimal to high lambda values, emphasizing the important influence of the combustion conditions on EF$_{\text{ROS}}$. For the PB, the EF$_{\text{ROS}}$ under $\lambda^{\text{opt}}(\lambda = 1.6)$ did not statistically differ from that under $\lambda^-$ ($\lambda \approx 1.3$) conditions for both primary and secondary emissions (Mann–Whitney, $p$ value $= 0.43$ and 0.20, respectively). When comparing the EF$_{\text{ROS}}$ under $\lambda^{\text{opt}}$ and $\lambda^-$ conditions with $\lambda^{++}$ ($2.7 < \lambda < 3.4$) conditions, primary EF$_{\text{ROS}}$ values under $\lambda^{\text{opt}}$ and $\lambda^-$ conditions were on average 7 and 3 times lower than that obtained under $\lambda^{++}$ conditions, respectively (Mann–Whitney, $p$ value $< 0.005$ for both cases). Aged EF$_{\text{ROS}}$ values under $\lambda^{\text{opt}}$ and $\lambda^-$ conditions were on average 8 and 5.5 times lower than obtained under $\lambda^{++}$ conditions, respectively (Mann–Whitney, $p$ value $= 0.02$ for both cases). In the MGB all the burns occurred at $2.0 < \lambda < 2.2$, leading to EF$_{\text{ROS}}$ in line with those from the PB between $\lambda^{\text{opt}}$ ($\lambda = 1.6$) and $\lambda^{++}$ (where $\lambda$ ranged from 2.7 to 3.4). The combustion in all stoves (PS, LWS1–4) exhibited higher $\lambda$ ($\lambda > 2.2$) due to a less controlled air supply leading to a lower combustion temperature and increased products of incomplete combustion (less efficient combustion). In this range of oxygen excess, all aged EF$_{\text{ROS}}$ were high but without any systematic trend with $\lambda$, suggesting that other param-

eters also influence PB-ROS emissions. We further revealed that this variability was related to the bulk OA emissions, implying that this variation is inherent to the combustion conditions.

Nonetheless, the PB-ROS content still varied by a factor of 2.6 on average for the same OA emission factor ($EF_{OA}$). We used a regression model on the data of SC and PAM chamber aging experiments to identify the different parameters that control the PB-ROS secondary formation and content in OA upon aging. This regression model showed that the PB-ROS contents in SOA (represented as $f_{ROS-SOA}$) depended significantly on all the aging parameters investigated, including the fractions of $m/z$ 44 and $m/z$ 43 in SOA, $f_{44-SOA}$ and $f_{43-SOA}$, respectively, the OH exposure, and the organic aerosol mass concentration. The greatest share of explained variability in $f_{ROS-SOA}$ was attributed to $f_{44-SOA}$, which indicates that the more oxygenated compounds are preferentially PB-ROS active compared to others. The OH exposure was the second most important parameter controlling the aerosol PB-ROS content under our conditions, where the anti-correlation between OH exposure and $f_{ROS-SOA}$ indicated that initially formed PB-ROS are prone to further reactions. The organic mass and $f_{43-SOA}$ exhibited a small, but statistically significant effect on $f_{ROS-SOA}$. In summary, the PB-ROS content seems to increase with the SOA oxidation state, which increases with OH exposure and decreases with the additional partitioning of semi-volatile components with lower PB-ROS content at higher OA concentrations, while further aging seems to result in a decay of PB-ROS. The comparison and evolution of PB-ROS with different combustion and aging conditions in this study could eventually provide a speedy assessment of potential health risks of wood combustion emissions from different combustion and aging conditions. However, a link between PB-ROS as measured with the DCFH method and oxidative stress in cell cultures and health effects needs still to be established.

*Data availability.* Data related to this article are available online at https://zenodo.org/record/1200236#.WqujTk2pUkk. TS7

**The Supplement related to this article is available online at https://doi.org/10.5194/acp-18-1-2018-supplement.**

*Competing interests.* The authors declare that they have no conflict of interest. TS8

*Acknowledgements.* This study was financially supported by the Swiss National Science Foundation (NRP 70 "Energy Turnaround"), the European Union's Horizon 2020 research and innovation programme through the EUROCHAMP-2020 Infrastructure Activity under grant agreement no. 730997, the Swiss National Science Foundation starting grant BSSGI0_155846, and the China Scholarship Council (CSC).

Edited by: Maria Cristina Facchini
Reviewed by: Rodney Weber and one anonymous referee

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

## Remarks from the language copy-editor

## Remarks from the typesetter