# Peer review of "Particle-bound reactive oxygen species (PB-ROS) emissions and formation pathways in residential wood smoke under different combustion and aging conditions"

_Atmospheric Chemistry and Physics, 2017_

## Referee Comment (RC1) · Anonymous Referee #1 · 21 Dec 2017

The manuscript presents novel data for the emission factors of reactive oxygen species (ROS) for several combustion devices commonly used in Europe. It also explores the effect of OH-induced ageing of smoke particles on their ROS content. The results of the ageing experiments highlight a complex behavior of redox-active compounds during ageing but the overall effect is a net increase with respect to purely primary emissions. Several interesting findings about the effect of burning conditions, burner operation, etc. are discussed in detail in the paper. The only aspect that, in this reviewer's opinion, has been neglected is about the employ of electrostatic precipitators

for the abatement of pollutants and ROS. In particular, the Authors' conclusion that "the use of the electrostatic precipitator had little effect on the primary and aged ROS emissions" is not fully consistent with the data reported in Table 2, where the ranged of EF-ROS before ESP (1758 – 2034 nmol Kg-1) is clearly smaller than after ESP (775 – 1098 nmol Kg-1). Beside this particular information that, once clarified, is worth to be mentioned in the Abstract, I have only one general comment. In my view, the paper lacks of an adequate description of the context for this research. For instance, the Introduction section in its current form misses important references to previous works done on ROS in aged combustion emissions but for fossil fuel (e.g., for diesel: Rattanavaraha et al., Atmos. Environ., 45, 3848-3855, 2011). References to past works on ROS associated to biomass burning aerosols but dealing with field measurements might be added (e.g., Verma et al., Environ. Sci. Technol., 49, 4646–4656, 2015). Finally, studies dealing with the cytotoxicity of ambient biomass burning aerosols collected in variable photochemical conditions can be quoted as well (e.g., Corsini et al., Sci Total Environ., v.587-588, 223-231, 2017). By expanding the context that the Authors feel appropriate for this laboratory research, its impact for the broader scientific community will become clearer.

---

## Referee Comment (RC2) · R. J. Weber (Referee) · 2 Jan 2018

This paper presents a very detailed analysis of reactive oxygen species (ROS) on aerosol particles produced from wood burning. The work systematically investigates differences in wood burners and burning conditions. Primary and secondary ROS is also studied. For what it is investigating, the paper is very clearly written and the methods and results are well supported. The paper was a pleasure to read. I have no major comments with the overall work, however, first it should be clarified exactly what species are thought to be measured with this assay (e.g., list them) and how, or what

evidence, or what is the basis for claiming throughout this paper that these species are associated with adverse health.

For example, a major overall issue that must be resolved is the conflation of different assays for measuring ROS and their links to adverse health. This paper uses the DCFH assay on collected particles to measure reactive oxygen species on or within the particles (or more specifically, components of the aerosol that will react with DCF, see comment above on specifying these). As far as I know of, there is no published study linking this measurement of ROS to adverse health effects, so it is not clear to me how these authors can relate this work to health effects, other than in a general way. In contrast there are a range of other assays that attempt to measure the ability of particles to consume reductants or generate ROS in vivo. Unfortunately these assays have also been referred to as measurements of ROS, but more recently are referred to as measurements of aerosol oxidative potential. These assays have been linked to adverse health responses, but are not what is being measured in this work. This paper tends to add to the confusion in the aerosol community on ROS by not precisely distinguishing the measurements presented here from other oxidative potential measurements. This needs to be rectified. The issue is discussed further below.

Specific comments

The abstract should be modified to clarify the type of ROS being investigated as the term ROS has many meanings. As an example, could change the first sentence to. . .

Wood combustion emissions can induce oxidative stress in the human respiratory tract caused by reactive oxygen species (ROS) on aerosol particles, that are emitted either directly or formed after oxidation in the atmosphere.

It would also be helpful to state somewhere in the abstract how ROS was measured, ie the DCFH assay. This would add much need clarity on what this paper means by the term ROS

Line 53-54, These adverse effects may be related to oxidative stress caused by free radicals induced by inhaled PM, which overwhelms the antioxidants in the body (Lobo et al., 2010). It seems this is the justification in this paper; that particle bound ROS measured with the DCFH assay is potentially associated with adverse health. But there is a very significant logical jump to assert particle bound ROS results in the physiological oxidative stress response (ie, as discussed in Lobo et al). For example, maybe all particle bound ROS is eliminated in the highly reducing environment of the lung lining fluid? Further more, this assay does not measure ROS induced by PM, it measures ROS on the particle as it exits in the ambient air.

Line 55 states: In turn, free radical formation may be due to reactive oxygen species (ROS) present in atmospheric aerosol, transition metals undergoing Fenton reactions, or redox cycling organic compounds like quinones.

Clarify where the free radical formation is occurring; within the particle in ambient air or once the particle is deposited and interacts with physiological species. This distinction is critical as they are both referred to as ROS, but are different processes and measured with different assays. To be clear, most health effects studies on ROS are not linked to the ROS being measured in this paper, they are linked to assays that measure ROS generation by physiological species interacting with aerosol species. This can be quantified through measuring ROS generated by cells exposed to particles and measuring loss of reductants or generation of oxidants in a simulated physiological environment. Example assays in the first case include measurements of cellular responses generating ROS measured with the DCFH probe, and in the second case acellular test assays such as DTT, GSH, Ascorbic Acid, where the loss of the reductant is measured (ie, DTT, GSH, AA) or various ways are used for detecting oxidants generated in the simulated environments. This is very different from the ROS measured in this paper.

If the authors know of references linking particle-bound DCFH-measured ROS (ie, what is reported in this paper) with adverse health endpoints, they should be included. If

there are none, maybe this should be noted as a motivation for needed research.

The authors could include references that specifically tested ROS formed in vivo (ie, not what is measured here) with health endpoints instead of providing only generic references on this subject (see ref. list at end).

Regarding metal ions, fenton reactions and quinones, does this assay detect these species or is this sentence saying that these species make ROS in the particle in the ambient atmosphere, which this assay measures. Please clarify. This is confusing because these species are mainly what drive the oxidative potential of aerosols measured with the assays noted above.

Ling 67 and throughout. It is really incorrect to call the measurements reported in this paper as oxidative potential. The measurements reported here are of the ROS associated with particles (i.e., $H_2O_2$, organic peroxides). Again, one must be very clear to distinguish this measurement from other assays, which largely measure a different property of the aerosol (ie, which should only be called oxidative potential), but unfortunately is also referred to as ROS.

It would be helpful if the authors included a discussion/summary of specifically what species the DCFH assay detects. (There are papers published on this).

Use of a PAM and smog chamber for aging experiments. In both cases please give specific OH concentrations the particles are exposed to so the reader has a sense of differences between these experiments and actual atmospheric aging (eg, ambient OH concentration is noted to be approx. 1E6 1/cm3). A brief discussion justifying this method for simulating actual aging processes should be included since this method comprises a major aspect of this paper and OH concentrations are likely very high and unrealistic. There will be chemical processes that do not occur in the actual ambient atmosphere that will occur in these experiments (ie, any radical-radical reactions pathways).

Line 207, this method is not measuring oxidative potential.

Ling 473-474 it states: This indicates that applying automatic combustion devices operated at optimum conditions, to achieve near-complete combustion, is most effective to minimize ROS emissions. This is a significant finding, but achieving near complete combustion also likely significantly reduces emissions of OA and BC. BC is known to be toxic. It might be important to also note this here.

Final line of paper: The comparison and evolution of ROS with different combustion and aging conditions in this study might be used for a speedy assessment of potential health risks of wood combustion emissions from different combustion and aging conditions. This assumes that the ROS associated with the particle is a health risk. Again, what evidence is there to support this?

Some References:

Delfino RJ, Staimer N, Tjoa T, Gillen DL, Schauer JJ, Shafer MM. Airway inflammation and oxidative potential of air pollutant particles in a pediatric asthma panel. Journal of Exposure Science and Environmental Epidemiology 2013;23:466-473.

Delfino RJ, Staimer N, Tjoa T, Arhami M, Polidori A, Gillen DL, George SC, Shafer MM, Schauer JJ, Sioutas C. Associations of Primary and Secondary Organic Aerosols With Airway and Systemic Inflammation in an Elderly Panel Cohort. Epidemiology 2010;21:892-902.

Zhang X, Staimer N, Gillen DL, Tjoa T, Schauer JJ, Shafer MM, Hasheminassab S, Pakbin P, Vaziri ND, Sioutas C, Delfino RJ. Associations of oxidative stress and inflammatory biomarkers with chemically-characterized air pollutant exposures in an elderly cohort. Environmental Research 2016;150:306-319.

Zhang X, Staimer N, Tjoa T, Gillen DL, Schauer JJ, Shafer MM, Hasheminassab S, Pakbin P, Longhurst J, Sioutas C, Delfino RJ. Associations between microvascular function and short-term exposure to traffic-related air pollution and particulate matter

oxidative potential. Environ Health 2016;15:81.

Strak M, Janssen NAH, Godri KJ, Gosens I, Mudway IS, Cassee FR, Lebret E, Kelly FJ, Harrison RM, Brunekreef B, Steenhof M, Hoek G. Respiratory Health Effects of Airborne Particulate Matter: The Role of Particle Size, Composition, and Oxidative Potential-The RAPTES Project. Environmental Health Perspectives 2012;120:1183-1189.

Steenhof M, Mudway IS, Gosens I, Hoek G, Godri KJ, Kelly FJ, Harrison RM, Pieters RHH, Cassee FR, Lebret E, Brunekreef BA, Strak M, Janssen NAH. Acute nasal pro-inflammatory response to air pollution depends on characteristics other than particle mass concentration or oxidative potential: the RAPTES project. Occupational and Environmental Medicine 2013;70:341-348.

Steenhof M, Janssen NAH, Strak M, Hoek G, Gosens I, Mudway IS, Kelly FJ, Harrison RM, Pieters RHH, Cassee FR, Brunekreef B. Air pollution exposure affects circulating white blood cell counts in healthy subjects: the role of particle composition, oxidative potential and gaseous pollutants - the RAPTES project. Inhalation Toxicology 2014;26:141-165.

Janssen NAH, Strak M, Yang A, Hellack B, Kelly FJ, Kuhlbusch TAJ, Harrison RM, Brunekreef B, Cassee FR, Steenhof M, Hoek G. Associations between three specific a-cellular measures of the oxidative potential of particulate matter and markers of acute airway and nasal inflammation in healthy volunteers. Occupational and Environmental Medicine 2015;72:49-56.

Weichenthal, S. A., D. L. Crouse, L. Pinault, K. Godri-Pollitt, W. Bavigne, G. Evans, A. v. Donkellar, R. V. Martin, and R. T. Burnett, Oxidative burden of fine particulate air pollution and risk of cause-specific mortality in the Canadian Census Health and Environment Cohort (CanCHEC), 2016, Envir. Res., 146, 92-99.

Weichenthal S, Lavigne E, Evans G, Pollitt K, Burnett RT. Ambient PM2.5 and risk of

emergency room visits for myocardial infarction: impact of regional PM2.5 oxidative potential: a case-crossover study. Environ Health 2016;15:46.

Weichenthal SA, Lavigne E, Evans GJ, Godri Pollitt KJ, Burnett RT. Fine Particulate Matter and Emergency Room Visits for Respiratory Illness. Effect Modification by Oxidative Potential. Am J Respir Crit Care Med 2016;194:577-586.

Yang A, Wang M, Eeftens M, Beelen R, Dons E, Leseman DLAC, Brunekreef B, Cassee FR, Janssen NAH, Hoek G. Spatial Variation and Land Use Regression Modeling of the Oxidative Potential of Fine Particles. Environmental Health Perspectives 2015;123:1187-1192.

Yang A, Janssen NA, Brunekreef B, Cassee FR, Hoek G, Gehring U. Children's respiratory health and oxidative potential of PM2.5: the PIAMA birth cohort study. Occup Environ Med 2016;73:154-160.

Tonne C, Yanosky JD, Beevers S, Wilkinson P, Kelly FJ. PM mass concentration and PM oxidative potential in relation to carotid intima-media thickness. Epidemiology 2012;23:486-494.

Bates JT, Weber RJ, Abrams J, Verma V, Fang T, Klein M, Strickland MJ, Sarnat SE, Chang HH, Mulholland JA, Tolbert PE, Russell AG. Reactive oxygen species generation linked to sources of atmospheric particulate matter and cardiorespiratory effects. Environmental Science & Technology 2015;49:13605-13612. PMID: 26457347.

Fang T, Verma V, Bates JT, Abrams J, Klein M, Strickland MJ, Sarnat SE, Chang HH, Mulholland JA, Tolbert PE, Russell AG, Weber RJ. Oxidative potential of ambient water-soluble PM2.5 in the southeastern United States: contrasts in sources and health associations between ascorbic acid (AA) and dithiothreitol (DTT) assays. Atmospheric Chemistry and Physics 2016;16:3865-3879.

Canova C, Minelli C, Dunster C, Kelly F, Shah PL, Caneja C, Tumilty MK, Burney P. PM10 oxidative properties and asthma and COPD. Epidemiology 2014;25:467-468.
Atkinson RW, Samoli E, Analitis A, Fuller GW, Green DC, Anderson HR, Purdie E, Durister C, Aitlhadj L, Kelly FJ, Mudway IS. Short-term associations between particle oxidative potential and daily mortality and hospital admissions in London. International Journal of Hygiene and Environmental Health 2016;219:566-572.

---

## Author Comment (AC1) · 7 Mar 2018

We thank the referees for their insightful questions and comments, which greatly helped to improve the quality of the paper. Our answers are listed in the following in red, after the reviewer's comments, which are in black. The modifications in the text are marked in yellow.

Anonymous Referee #1

General comments: The manuscript presents novel data for the emission factors of

reactive oxygen species (Zhao and Hopke) for several combustion devices commonly used in Europe. It also explores the effect of OH-induced ageing of smoke particles on their ROS content. The results of the ageing experiments highlight a complex behavior of redox-active compounds during ageing but the overall effect is a net increase with respect to purely primary emissions. Several interesting findings about the effect of burning conditions, burner operation, etc. are discussed in detail in the paper. The only aspect that, in this reviewer's opinion, has been neglected is about the employ of electrostatic precipitators for the abatement of pollutants and ROS. In particular, the Authors' conclusion that "the use of the electrostatic precipitator had little effect on the primary and aged ROS emissions" is not fully consistent with the data reported in Table 2, where the ranged of EF-ROS before ESP (1758 – 2034 nmol Kg-1) is clearly smaller than after ESP (775 – 1098 nmol Kg-1). Beside this particular information that, once clarified, is worth to be mentioned in the Abstract, I have only one general comment. In my view, the paper lacks of an adequate description of the context for this research. For instance, the Introduction section in its current form misses important references to previous works done on ROS in aged combustion emissions but for fossil fuel (e.g., for diesel: Rattanavaraha et al., Atmos. Environ., 45, 3848-3855, 2011). References to past works on ROS associated to biomass burning aerosols but dealing with field measurements might be added (e.g., Verma et al., Environ. Sci. Technol., 49, 4646–4656, 2015). Finally, studies dealing with the cytotoxicity of ambient biomass burning aerosols collected in variable photochemical conditions can be quoted as well (e.g., Corsini et al., Sci Total Environ., v.587-588, 223-231, 2017). By expanding the context that the Authors feel appropriate for this laboratory research, its impact for the broader scientific community will become clearer. Indeed, the emissions after the electrostatic precipitator are $\sim$ 1.5 times lower but still within the burn-to-burn variability. We changed the sentence to "The use of the electrostatic precipitator decreased primary and aged ROS emissions on average by a factor of $\sim$1.5, however, these differences are not statistically significant (Mann-Whitney p-value = 0.12 for both primary and aged emissions) and are within the burn-to-burn variability." In the abstract we added: "The

use of an electrostatic precipitator decreased the primary and aged ROS emissions by a factor of ~1.5 which is however still within the burn-to-burn variability." To better link our work with previous work of primary and aged natural and specific source emissions, we included in the introduction the references mentioned by the reviewer and added some others in our response to Rodney Weber's comments.

R. J. Weber (Referee#2)

General comments: This paper presents a very detailed analysis of reactive oxygen species (ROS) on aerosol particles produced from wood burning. The work systematically investigates differences in wood burners and burning conditions. Primary and secondary ROS is also studied. For what it is investigating, the paper is very clearly written and the methods and results are well supported. The paper was a pleasure to read. I have no major comments with the overall work, however, first it should be clarified exactly what species are thought to be measured with this assay (e.g., list them) and how, or what evidence, or what is the basis for claiming throughout this paper that these species are associated with adverse health. For example, a major overall issue that must be resolved is the conflation of different assays for measuring ROS and their links to adverse health. This paper uses the DCFH assay on collected particles to measure reactive oxygen species on or within the particles (or more specifically, components of the aerosol that will react with DCF, see comment above on specifying these). As far as I know of, there is no published study linking this measurement of ROS to adverse health effects, so it is not clear to me how these authors can relate this work to health effects, other than in a general way. In contrast there are a range of other assays that attempt to measure the ability of particles to consume reductants or generate ROS in vivo. Unfortunately these assays have also been referred to as measurements of ROS, but more recently are referred to as measurements of aerosol oxidative potential. These assays have been linked to adverse health responses, but are not what is being measured in this work. This paper tends to add to the confusion in the aerosol community on ROS by not precisely distinguishing the measurements presented here from other oxidative potential measurements. This needs to be rectified. The issue is discussed further below.

This is a very valuable point of discussion. We clarified this in our answers to the more detailed comments of the reviewer raised below.

Specific comments:

The abstract should be modified to clarify the type of ROS being investigated as the term ROS has many meanings. As an example, could change the first sentence to...

Wood combustion emissions can induce oxidative stress in the human respiratory tract caused by reactive oxygen species (ROS) on aerosol particles, that are emitted either directly or formed after oxidation in the atmosphere.

We modified the abstract according to the reviewer's suggestion.

It would also be helpful to state somewhere in the abstract how ROS was measured, ie the DCFH assay. This would add much need clarity on what this paper means by the term ROS rinter-friendly version We mention now the DCFH assay in the abstract.

We added the following (marked in yellow): The PB-ROS content as well as the chemical properties of the aerosols were quantified by a novel ROS analyzer using the DCFH (2',7'-dichlorofluorescin) assay.....

Line 53-54, These adverse effects may be related to oxidative stress caused by free radicals induced by inhaled PM, which overwhelms the antioxidants in the body (Lobo et al., 2010). It seems this is the justification in this paper; that particle bound ROS measured with the DCFH assay is potentially associated with adverse health. But there is a very significant logical jump to assert particle bound ROS results in the physiological oxidative stress response (ie, as discussed in Lobo et al). For example, maybe all particle bound ROS is eliminated in the highly reducing environment of the lung lining fluid? Furthermore, this assay does not measure ROS induced by PM, it measures ROS on the particle as it exists in the ambient air.

This sentence is a general statement on a prevalent hypothesis how adverse effects of aerosols may be induced. We believe that the reviewer thinks along the same line as he states in Verma et al., ES&T 2015: "Ambient aerosols may lead to oxidative stress by transporting oxidants on particles into the respiratory system, or introducing aerosol components that are capable of catalytically generating reactive oxygen species (ROS) both in vivo and in vitro". While our method could be a measure of the particle-bound ROS (PB-ROS) transported into the respiratory system, other assays like dithiothreitol (DTT) and cellular probes rather measure ROS generation in cells or artificial lung fluids. It can however not be excluded that the latter methods also respond to some extent to PB-ROS. The reviewer is completely right that the measurement of particle bound ROS does not provide a measure of the full induced oxidative stress. Indeed, there are no studies of ambient or ambient-like aerosols that provide a link between PB-ROS and inflammatory biomarkers. Still we believe it is useful to measure PB-ROS as part of the inflammatory causes. There is the study by Lakey et al. (2016) who modeled the ROS produced in the human respiratory tract upon inhalation of PM and showed that the OH production rate from SOA can be as high as the $H_2O_2$ production rate from trace metals. Summarizing, we believe that a combination of different assays is needed to disentangle the various causes of inflammatory processes.

We clarify this in the introduction (see text after next comment).

Line 55 states: In turn, free radical formation may be due to reactive oxygen species (ROS) present in atmospheric aerosol, transition metals undergoing Fenton reactions, or redox cycling organic compounds like quinones. Clarify where the free radical formation is occurring; within the particle in ambient air or once the particle is deposited and interacts with physiological species. This distinction is critical as they are both referred to as ROS, but are different processes and measured with different assays. To be clear, most health effects studies on ROS are not linked to the ROS being measured in this paper, they are linked to assays that measure ROS generation by physiological species interacting with aerosol species. This can be quantified through measuring ROS generated by cells exposed to particles and measuring loss of reductants or generation of oxidants in a simulated physiological environment. Example assays in the first case include measurements of cellular responses generating ROS measured with the DCFH probe, and in the second case acellular test assays such as DTT, GSH, Ascorbic Acid, where the loss of the reductant is measured (ie, DTT, GSH, AA) or various ways are used for detecting oxidants generated in the simulated environments. This is very different from the ROS measured in this paper. If the authors know of references linking particle-bound DCFH-measured ROS (ie, what is reported in this paper) with adverse health endpoints, they should be included. If there are none, maybe this should be noted as a motivation for needed research.

We added the clarification where the free radical formation is occurring. Furthermore, we mention that there is no study linking the DCFH assay of ambient aerosol with inflammatory markers and provide our hypothesis to use the DCFH assay. The modified text reads now: This may happen via two pathways. 1) Particles may contain reactive oxygen species (particle-bound ROS, PB-ROS, exogenous), which act as oxidants in the biological system; 2) particles contain transition metals or organic compounds like quinones, which generate reactive oxygen species by interaction with physiological species undergoing Fenton reactions and redox cycling. For the measurement of the latter property several assays have been introduced, where the loss of reductants like dithiothreitol (DTT), glutathione (GSH) or ascorbic acid (AA) is measured (Cho et al., 2005; Verma et al., 2012; Charrier and Anastasio, 2012; Verma et al., 2015; Fang et al., 2016; Weber et al., 2018; Mudway et al., 2004; Li et al., 2003; Fang et al., 2016). Also cellular tests with an ROS probe have been developed to measure ROS induced by aerosols in a biotic system (Landreman et al., 2008; Zhang et al., 2016; Tuet et al., 2017). The DCFH assay has been shown to be sensitive to a broad range of organic peroxides, alkyl peroxide radicals, and hypochlorite, but not to components that are known to induce redox cycling (i.e., metal ions and quinones) (Venkatachari and Hopke, 2008; Wang et al., 2011; King and Weber, 2013; Fuller et al., 2014b; Zhou et al., 2018). The DCFH assay has fast response rates and a linear response to varying ROS concentrations, for which reason it was applied as a suitable measure for the particle bound ROS concentration (Zhou et al., 1997; Venkatachari and Hopke, 2008; King and Weber, 2013, Zhou et al., 2018). Several studies tried to establish links between such measurements and specific inflammatory biomarkers of oxidative stress in cell cultures or human subjects. For example Delfino et al. (2010; 2013) found that macrophage ROS generation from PM2.5 was significantly positively associated with nitric oxide exhaled from elderly subjects and school children with persistent asthma. Janssen et al. (2015) reported a significant association between exhaled nitric oxide and increases in interleukin-6 in nasal lavage and the ROS generation measured by DTT and AA. Others found correlations between DTT activity and emergency department visits for asthma/wheezing and congestive heart failure (Bates et al., 2015; Fang et al., 2016). We are not aware of a study that relates measurements of ambient PB-ROS by DCFH with inflammatory biomarkers. However, it was shown that fresh SOA can release OH and H2O2 when dissolved in water (Wang et al., 2011; Tong et al. 2016). This phenomenon was attributed to labile peroxides contained in SOA, which were shown do decay with a rather short lifetime of less than an hour (Krapf et al., 2016). Lakey et al. (2016) modeled the ROS produced in the human respiratory tract upon inhalation of PM and showed that the OH production rate from SOA can be as high as the H2O2 production rate from trace metals. This indicates that PB-ROS might also lead to oxidative stress. Obviously, further research is needed to establish a link between PB-ROS and health effects. Based on these considerations we performed a study with a DCFH assay to focus on exogenous ROS formed by wood combustion exhaust and during its atmospheric aging. For this purpose, a suite of SC and PAM experiments were conducted. As different types of wood, combustion appliances and combustion conditions result in varying levels of emissions (Johansson et al., 2004; Schmidl et al., 2011; Fitzpatrick et al., 2007; Heringa et al, 2011), eight wood combustion devices with variable combustion conditions were tested. Primary and aged biomass smoke generated under different combustion and aging conditions were characterized by an online ROS analyzer based on the DCFH assay coupled with

an aerosol collector. Observations from this study provide more detailed evidence of the influence of combustion technology on the PB-ROS of the emitted PM compared to a previous similar study (Miljevic et al., 2010). We also show the variation of the PB-ROS content from primary and aged aerosols under different operation conditions. Further, the contribution of reactive oxygen species to aged organic aerosol generated with different aging tools was investigated to clarify the PB-ROS formation potential upon photo-oxidation. Results from these experiments may be directly compared with ambient measurements.

The authors could include references that specifically tested ROS formed in vivo (ie, not what is measured here) with health endpoints instead of providing only generic references on this subject (see ref. list at end).

We have added such references. See text above.

Regarding metal ions, fenton reactions and quinones, does this assay detect these species or is this sentence saying that these species make ROS in the particle in the ambient atmosphere, which this assay measures. Please clarify. This is confusing because these species are mainly what drive the oxidative potential of aerosols measured with the assays noted above.

This is not what the sentence should say. The DCFH assay does neither detect metal ions nor quinones. Our tests showed that Fe3+ had no influence on the ROS signal while soluble Fe2+ reduced it slightly when present at high concentrations and anthraquinone barely reacted (Zhou et al., 2018). DCFH measures only the particle-bound organic peroxides. Fenton reactions and quinones are measured by other assays like DTT.

We changed the sentence and clarified it as given in the text above.

Ling 67 and throughout. It is really incorrect to call the measurements reported in this paper as oxidative potential. The measurements reported here are of the ROS asso-

ciated with particles (i.e., H2O2, organic peroxides). Again, one must be very clear to distinguish this measurement from other assays, which largely measure a different property of the aerosol (ie, which should only be called oxidative potential), but unfortunately is also referred to as ROS. We changed the wording and call the measurements now particle-bound ROS (PB-ROS).

It would be helpful if the authors included a discussion/summary of specifically what species the DCFH assay detects. (There are papers published on this).

We mention now the species detected by the DCFH assay, see modified introduction above.

Use of a PAM and smog chamber for aging experiments. In both cases please give specific OH concentrations the particles are exposed to so the reader has a sense of differences between these experiments and actual atmospheric aging (eg, ambient OH concentration is noted to be approx. 1E6 1/cm3). A brief discussion justifying this method for simulating actual aging processes should be included since this method comprises a major aspect of this paper and OH concentrations are likely very high and unrealistic. There will be chemical processes that do not occur in the actual ambient atmosphere that will occur in these experiments (ie, any radical-radical reactions pathways).

The information on the OH exposure in the PAM and in the smog chamber is already given in Sect. 2.3.2. In the PAM chamber, all the experiments were conducted under OH exposures of $(1.1\text{-}2.0)\times10^8$ molecules cm-3 h which corresponds to $\sim$ 4.5-8 days of aging in the ambient atmosphere. In the two smog chambers (PSI-MSC PSI-SSC) we performed experiments with an OH exposure of $2.6\text{-}4.8\times10^7$ molecules cm-3 h ($\sim$1-2 days of aging in the ambient atmosphere) up to $4.0\times10^8$ molecules cm-3 h, which is equivalent to $\sim$17 days of ambient aging.

In Section 3.3 we discuss the influence of aging conditions PB-ROS formation in detail and in the paragraph "comparison between SC- and PAM chamber data" we demonstrate that the PAM results are in line with the smog chamber data.

Consequently, we think that there is already sufficient information on the reviewer's comment in the manuscript.

Line 207, this method is not measuring oxidative potential.

We changed the wording to particle-bound ROS.

Ling 473-474 it states: This indicates that applying automatic combustion devices operated at optimum conditions, to achieve near-complete combustion, is most effective to minimize ROS emissions. This is a significant finding, but achieving near complete combustion also likely significantly reduces emissions of OA and BC. BC is known to be toxic. It might be important to also note this here.

Indeed, the combustion phases and conditions also affect POA, SOA and BC. A detailed presentation of these effects will be subject of another paper. A combined view will be given there.

We changed the sentence to: This indicates that applying automatic combustion devices operated at optimum conditions, to achieve near-complete combustion, is most effective to minimize ROS emissions, but also POA, SOA and BC.

Final line of paper: The comparison and evolution of ROS with different combustion and aging conditions in this study might be used for a speedy assessment of potential health risks of wood combustion emissions from different combustion and aging conditions. This assumes that the ROS associated with the particle is a health risk. Again, what evidence is there to support this?

[revised manuscript text omitted]

Zhao, J., and Hopke, P. K.: Concentration of Reactive Oxygen Species (ROS) in Mainstream and Sidestream Cigarette Smoke, Aerosol Science and Technology, 46, 191-197, http://dx.doi.org/10.1080/02786826.2011.617795, 2012. Zhou, J., Bruns, E. A., Zotter, P., Stefenelli, G., Prévôt, A. S. H., Baltensperger, U., El-Haddad, I., and Dommen, J.: Development, characterization and first deployment of an improved online reactive oxygen species analyzer, Atmos. Meas. Tech., 11, 65-80, http://dx.doi.org/10.5194/amt-11-65-2018, 2018. Zhou, M., Diwu, Z., Panchuk-Voloshina, N., and Haugland, R. P.: A stable nonfluorescent derivative of resorufin for the fluorometric determination of trace hydrogen peroxide: applications in detecting the activity of phagocyte NADPH oxidase and other oxidases, Anal. biochem., 253, 162-168, http://dx.doi.org/10.1006/abio.1997.2391, 1997.